# Development and Characterization of ZnO Piezoelectric Thin Film Sensors on GH4169 Superalloy Steel Substrate by Magnetron Sputtering

**DOI:** 10.3390/mi13030390

**Published:** 2022-02-28

**Authors:** Guowei Mo, Yunxian Cui, Junwei Yin, Pengfei Gao

**Affiliations:** Mechanical and Electronic Engineering, College of Mechanical Engineering, Dalian Jiaotong University, Dalian 116024, China; moguowei00@163.com (G.M.); yjwzx@djtu.edu.cn (J.Y.); gaopengfei198935@163.com (P.G.)

**Keywords:** ZnO piezoelectric thin films, GH4169 superalloy steel substrate, piezoelectric sensors, static and dynamic characteristics, stability

## Abstract

At present, piezoelectric sensors are primarily applied in health monitoring areas. They may fall off owing to the adhesive’s durability, and even damage the monitored equipment. In this paper, a piezoelectric film sensor (PFS) based on a positive piezoelectric effect (PPE) is presented and a ZnO film is deposited on a GH4169 superalloy steel (GSS) substrate using magnetron sputtering. The microstructure and micrograph of ZnO piezoelectric thin films were analyzed by an X-ray diffractometer (XRD), energy dispersive spectrometer (EDS), scanning electron microscope (SEM), and atomic force microscope (AFM). The results showed that the surface morphology was dense and uniform and had a good c-axis-preferred orientation. According to the test results of five piezoelectric sensors, the average value of the longitudinal piezoelectric coefficient was 1.36 pC/N, and the average value of the static calibration sensitivity was 19.77 mV/N. We selected the sensor whose parameters are closest to the average value for the dynamic test experiment and we drew the output voltage response curve of the piezoelectric film sensor under different loads. The measurement error was 4.03% when repeating the experiment six times. The research achievements reveal the excellent performance of the piezoelectric film sensor directly deposited on a GH4169 superalloy steel substrate. This method can reduce measurement error caused by the adhesive and reduce the risk of falling off caused by the aging of the adhesive, which provides a basis for the research of smart bolts and guarantees a better application in structural health monitoring (SHM).

## 1. Introduction

Piezoelectric sensors have been widely applied in structural health monitoring to improve the safety and reliability of structures and to reduce life-cycle costs [1,2,3]. For this reason, the study of piezoelectric sensors including PZT [4,5], ALN [6,7,8], PVDF [9,10], ZnO [11,12], etc., has become a hot topic. The sensors are usually mounted on the surface of a structure or embedded into the structure to monitor its working state. Many research results have been achieved in structural health monitoring, such as bolt health monitoring [13], old planes real-time monitoring [14], flexible structure health monitoring [15], and composite structure health monitoring [16,17].

Among the existing piezoelectric materials, ZnO has attracted huge attention. It has the advantages of being a third-generation semiconductor, including excellent piezoelectric property, high piezoelectric coupling coefficient, optical and electrochemical properties, cheap manufacturing cost, and environment-friendly fabrication process [18,19,20]. Based on different working principles, it is also widely used in photoelectric sensors [21], nanogenerators [22], acceleration sensors [23], and pressure sensors [24]. ZnO piezo-layers have been deposited on the substrate by many methods, such as chemical vapor deposition [25], pulsed laser deposition [26], sol-gel processing [27], and magnetron sputtering [28]. In the structural health monitoring field, a ZnO piezoelectric film sensor monitors pressure and acceleration mainly by piezoelectric characteristics. The working principle is to convert mechanical energy into electrical energy by using the positive piezoelectric effect. Don L. et al. fabricated two kinds of piezoelectric accelerometers by single-target RF sputtering and derived a complex coupling relationship between the resonant frequency and device sensitivity. By comparison, the simple cantilever accelerometer has a higher sensitivity than another at the same resonant frequencies [29]. In order to measure acceleration of small and lightweight devices, Hyun et al. reported a miniaturized ZnO piezoelectric film accelerometer, which was deposited on a copper wafer by the refresh hydrothermal synthesis. The sensitivity was up to 37.7 Pa/g, which was about 30 times larger than the previous result. The performance evaluation results showed that the accelerometer’s output current rose linearly with the acceleration [30,31]. In addition to the application of ZnO films in piezoelectric accelerometers, many researchers have also conducted a lot of research on the application of ZnO piezoelectric films in pressure sensors. Andreas et al. presented a ZnO piezoelectric film sensor that was fabricated on the polyimide membrane by using standard MEMS technologies; then, the polyimide sheet was thermally bonded to an already micromachined silicon wafer. The flow sensor has the advantage of measuring pressure across a fluidic restriction without contacting liquid [32]. Chang et al. designed a ZnO thin film pressure sensor running at high temperatures and obtained structures similar to silicon-on-insulator ones by growing ZnO thin films on SiO_2_/(100) Si substrates. The response value of the pressure sensor was proportional to pressure, greater than 8 mΩ/psi, and the authors obtained the change of gas pressure by measuring the change of resistivity [33]. Based on these, Karina Jeronimo et al. [34] discussed the effect of ZnO fillers for piezoelectric properties and Lee et al. [11] analyzed the influence of the annealing temperature on enhancing the output voltage of a force sensor.

The above research results have shown that ZnO piezoelectric film sensors have a mature application in the MEMS field, and it has also revealed that ZnO has good compatibility with Si, SiO_2_, Cu, etc. These properties provide a research basis for the application of ZnO piezoelectric film sensors in the field of structural health monitoring. Normally, a piezoelectric film sensor is fixed on the monitored mechanism by an adhesive to diagnose the health status of the target, such as a pressure liquid flow [32], breath and heart rate [35], gas pressure [33], etc. The research background of this paper is the fabrication of smart bolts. In order to monitor the working condition of bolts, Huo et al. proposed a piezoceramic-transducer-enabled time-reversal method to monitor the load change of a rock bolt by pasting a PZT patch. A PZT patch generated stress waves received by a smart washer, and the signal changes of the preload were analyzed by the time-reversal technique [36]. Similarly, Shao et al. estimated the bolt preload by the frequency shift method of the piezoelectric impedance. The PZT patch was pasted on the bolt head by an adhesive to diagnose the bolt looseness status in actual working conditions [37]. However, due to the adhesive’s durability, there is a risk that the sensors will fall off with the vibration of the monitoring equipment. Moreover, the use of an adhesive between the sensors and the monitored mechanism increases the measurement error. To solve this problem, Chang et al. fabricated a wind-power generator based on ZnO piezoelectric films by an RF magnetron sputtering system. ZnO films were deposited on the stainless-steel substrate at a temperature of 300 °C, with an output power up to 1.0 μW/cm^2^ [38]. Based on the development of these technologies, the direct integration of piezoelectric sensors on the bolt can be designed and applied for on-site monitoring of bolt preloads. The bolt we studied is applied to aviation turbine engines, the material was GH4169 superalloy steel. To date, researchers have not reported on the reliability of a ZnO piezoelectric film sensor deposited on a GH4169 superalloy steel substrate.

In this paper, we deposited a ZnO piezoelectric film sensor on a GH4169 superalloy steel substrate by magnetron sputtering and achieved the coupling relationship of piezoelectricity and mechanical stress. The sensor structure was designed according to the size of the bolt head, NiCr was selected as the electrode, and SiO_2_ as the protective layer. The surface morphology and crystal orientation of ZnO thin films were analyzed using EDS, XRD, SEM, and AFM. To obtain the conversion relationship of strain-to-electricity energy, the electrical performance of the ZnO piezoelectric film sensor was examined using a dynamic testing system. The properties considered in the evaluation of the ZnO piezoelectric film sensor performance included static and dynamic characteristics analysis, piezoelectric coefficient *d*_33_, sensitivity, and stability. This study provides a theoretical basis for the further research of smart bolts and promotes the development of superalloy smart bolts for the structural health monitoring field.

## 2. Working Principle and Structural Design

### 2.1. Working Principle

Figure 1 shows the working principle of the piezoelectric film sensor. The positive piezoelectric effect is used to convert mechanical energy into electrical energy. Generally, the electric charges are neutral without external force conditions, as shown in Figure 1a. However, when an external force *F* acts on the top, the charges shift from the centers of the ZnO films, which leads to forming positive and negative electrodes, as shown in Figure 1b. The sensor can be equated to a capacitor under the above conditions, and $V_{\mathrm{out}}$ is the output voltage between the upper and lower electrodes. The change of the mechanical structure force can be monitored by the change of the voltage value.

By analyzing the elastic mechanism of a piezoelectric film, the stress σ of the circular piezoelectric layer is
(1)$$\sigma =\frac{4F}{\pi d^{2}}$$
where F is the external force applied to the piezoelectric thin films and d is the diameter.

Based on the piezoelectric effect, the functional relationship between the induced charge density q and stress σ is [39]:(2)$$\left(\begin{array}{l} q_{1}\\ q_{2}\\ q_{3} \end{array}\right)=\left(\begin{array}{cccccc} d_{11} & d_{12} & d_{13} & d_{14} & d_{15} & d_{16}\\ d_{21} & d_{22} & d_{23} & d_{24} & d_{25} & d_{26}\\ d_{31} & d_{32} & d_{33} & d_{34} & d_{35} & d_{36} \end{array}\right)\left(\begin{array}{c} \sigma_{1}\\ \sigma_{2}\\ \sigma_{3}\\ \sigma_{4}\\ \sigma_{5}\\ \sigma_{6} \end{array}\right)$$
where $q_{1}$, $q_{2}$, and $q_{3}$ are piezoelectric induced charge densities on the ZnO film surface, $d_{11}$ to $d_{36}$ are piezoelectric coefficients, $\sigma_{1}$, $\sigma_{2}$, $\sigma_{3}$ are the axial stress of each plane and $\sigma_{4}$, $\sigma_{5}$, $\sigma_{6}$ are tangential stresses.

There are five nonzero elements of ZnO materials, namely, $d_{31}=d_{32}$, $d_{16}=d_{24}$ and $d_{33}$. As a first-order approximation for flat membrane structures, after ignoring $\sigma_{2}$, $\sigma_{3}$, $\sigma_{4}$, $\sigma_{5}$, $\sigma_{6}$, piezoelectric Equation (2) is simplified to:(3)$$\left(\begin{array}{l} q_{1}\\ q_{2}\\ q_{3} \end{array}\right)=\left(\begin{array}{cccccc} 0 & 0 & 0 & 0 & d_{15} & 0\\ 0 & 0 & 0 & d_{15} & 0 & 0\\ d_{31} & d_{31} & d_{33} & 0 & 0 & 0 \end{array}\right)\left(\begin{array}{c} \sigma_{1}\\ 0\\ 0\\ 0\\ 0\\ 0 \end{array}\right)$$

It can be obtained from Equation (3)
(4)$$q_{1}=0,\;q_{2}=0,\;q_{3}=d_{31}\sigma_{1}$$

By substituting Equation (1) into (4), $q_{3}$ can be obtained:(5)$$q_{3}=d_{31}\frac{4F}{\pi d^{2}}$$

In Equation (5), d is constant. When the external force F is constant, it can be obtained that the charge density $q_{3}$ is directly proportional to the piezoelectric coefficient $d_{31}$. When the piezoelectric coefficient $d_{31}$ is constant, it can be obtained that the charge density $q_{3}$ is directly proportional to the external force F.

### 2.2. Structural Design

We selected NiCr as electrode material because of its stable chemical stability and its difficulty to be oxidized in air. In addition, the process of preparing a NiCr film in our laboratory is mature. Typical insulating materials include Al_2_O_3_, Si_3_N_4_, SiO_2_, etc. SiO_2_ has a simple preparation process and low cost, its resistance value can reach 10^9^ Ω, it has excellent corrosion resistance and stability, and the maximum withstand temperature is up to 1000 °C [40]. Therefore, SiO_2_ was selected as the insulating film and protective film.

The piezoelectric film sensor was designed according to the principle of the positive piezoelectric effect, and was deposited on a GH4169 superalloy steel substrate, including a SiO_2_ insulating layer, NiCr electrode, ZnO piezoelectric layer, NiCr electrode and SiO_2_ protective layer, as shown in Figure 2a. To facilitate the test, the leakage size of the designed electrode was about 2 mm × 2 mm after the deposition of the protective layer, as shown in Figure 2b.

## 3. Experimental Setups and Processes

The magnetron sputtering system used in this paper was the JGP450B multitarget magnetron sputtering system designed by the Shenyang scientific instrument development center of the Chinese Academy of Sciences. All films were deposited without a substrate bias, the distance between the target and the substrate was 65 mm. The size of the GH4169 superalloy steel substrates was 18 × 18 × 0.6 mm and its properties are presented in Table 1. The sputtering parameters of the SiO_2_, ZnO, and NiCr films are shown in Table 2.

The principle diagram of RF magnetron sputtering is shown in Figure 3a. In the fabrication of ZnO Films, the Zn target was used as cathode, and the substrate and mask were used as anode. After the vacuum chamber was pumped to a certain vacuum degree by a molecular pump, Ar gas filled the chamber as sputtering gas. When a high number of kilovolts was applied to the target, the kinetic energy of the Zn atoms splashed can reach tens of electronvolts. When the kinetic energy was 2–3 orders of magnitude higher, the Zn target was always cleaned and activated in the plasma area during the film-forming process and the ZnO with weak adhesion can be removed. In order to improve the quality of the ZnO films and the consistency of the c-axis orientation, O_2_ needs to fill the vacuum chamber as the reaction gas. The simplified diagram of the RF magnetron sputtering system is shown in Figure 3b.

### 3.1. Fabrication of the Piezoelectric Film Sensor

Figure 4 shows the primary fabrication process of the piezoelectric film sensor on the GH4169 superalloy steel substrate by magnetron sputtering. The main steps were: (a) polishing the substrate to a mirror surface, washed successively with acetone, absolute ethanol, and deionized water for 15min, dried with nitrogen, and sputtered for 30 min to thoroughly clean the surface; (b) growing a SiO_2_ insulating layer without the stainless-steel mask plate on the whole plane of the substrate, growth process parameters are shown in Table 2; (c) depositing a NiCr bottom electrode layer on the SiO_2_ insulating layer with the stainless-steel mask plate as shown in Figure 5a, (d) growing the ZnO piezoelectric thin film by magnetron sputtering with the stainless-steel mask plate as shown in Figure 5b; (e) preparing the NiCr top electrode film on the ZnO piezoelectric layer with the stainless-steel mask plate as shown in Figure 5c; (f) growing the SiO_2_ protective layer film at last with the stainless-steel mask plate as shown in Figure 5d.

#### 3.1.1. Fabrication of the SiO_2_ Thin Film Layer

It is necessary to make a SiO_2_ insulating layer at the bottom of the sensor to prevent charge leakage because of the substrate conductivity of GH4169 superalloy steel. Similarly, we deposited a SiO_2_ protective layer on the top to prevent the sensor from being polluted after the sensor was sputtered. The thickness of sputtered SiO_2_ reached 2.7 μm, and the resistance can reach 10^8^ Ω and meet the insulation requirements. Figure 4b shows the insulating layer and Figure 4f shows the protective layer. The sputtering parameters of the SiO_2_ thin film are shown in Table 2.

#### 3.1.2. Fabrication of the NiCr Electrode Layer

We installed the stainless-steel mask plate (Figure 5a) on the fixture (Figure 5f) after the deposition of the SiO_2_ insulating layer. The target was replaced by NiCr, and the NiCr bottom electrode film with a thickness of 200 nm was deposited. Similarly, we installed the stainless-steel mask plate (Figure 5c) on the fixture (Figure 5f) after preparing the ZnO piezoelectric film. The target was replaced again to prepare a NiCr top electrode film with a thickness of 200 nm. The deposition parameters of the NiCr thin film are shown in Table 2.

#### 3.1.3. Fabrication of the ZnO Piezoelectric Layer

After the deposition of the NiCr bottom electrode, the target was replaced by Zn (99.999%), then we installed the stainless-steel mask plate (Figure 5b) on the fixture (Figure 5f) to sputter the ZnO piezoelectric film. The films were sputtered with a thickness of 1.0 μm. The sputtering process parameters have been optimized in our previous research. They are shown in Table 2.

A sample fixture (Figure 5f) was designed to ensure that the concentricity of each film layer met the requirements. The fixture body was installed at the position marked substrate in Figure 3b. It can prepare six samples for each group at the same time. The positioning accuracy depends on the six positioning columns and sample positioning plates. The fixture installation is shown in Figure 5f. Figure 5e shows that the sizes of the six stations were 18 × 18 × 0.5 mm. Due to the different shapes of each layer film, we needed to cover the unnecessary parts by a stainless-steel mask plate. Figure 5a–d shows the shape and key dimensions of the stainless-steel mask plate used with the fixture; the thickness of SMMP was 0.5 mm.

### 3.2. Testing Device of the Piezoelectric Film Sensor

All sensors must be calibrated to determine their essential characteristics after being manufactured and assembled. There are important performance indicators for piezoelectric film sensors, such as static characteristics, dynamic characteristics, and stability. Therefore, it is imperative to calibrate the developed piezoelectric film sensors.

The static test adopts the principle of the quasi-static method. According to the positive piezoelectric effect, a low-frequency alternating force whose frequency is far lower than the resonant frequency of the vibrator was applied to the piezoelectric vibrator to produce an alternating charge. When the vibrator meets the electrical short-circuit boundary condition without the action of external electric field and only bears force parallel to the polarization direction, the piezoelectric equation [42] can be simplified as follows:(6)$$D_{3}=d_{33}T_{3}d_{33}=\frac{D_{3}}{T_{3}}=\frac{Q}{F}$$
where $D_{33}$ is potential shift component, C/m^2^; $T_{3}$ is the longitudinal stress, N/m^2^; $d_{33}$ is the longitudinal piezoelectric strain constant, C/N or M/V; Q is the piezoelectric charge released by an oscillator, C; F is the longitudinal low-frequency alternating force, N.

If a measured vibrator is mechanically connected in series with a known comparison vibrator, a low-frequency alternating force is generated by an electromagnetic driver in a force application device and applied to the vibrator, as shown in Figure 6a.

Then, the piezoelectric charge $Q_{1}$ released by the measured oscillator establishes a voltage $V_{1}$ on its parallel capacitor $C_{1}$, while the piezoelectric charge $Q_{2}$ released by the comparison oscillator establishes a voltage $V_{2}$ on $C_{2}$. The following can be obtained from Equation (6):(7)$$\begin{array}{l} d_{33}^{\left(1\right)}=\frac{C_{1}V_{1}}{F}\\ d_{33}^{\left(2\right)}=\frac{C_{2}V_{2}}{F} \end{array}\}$$
where: $C_{1}=C_{2}>100C^{T}$ (oscillator free capacitance).

Equation (7) can be further reduced to:(8)$$d_{33}^{\left(1\right)}=\frac{V_{1}}{V_{2}}d_{33}^{\left(2\right)}$$

In Equation (8), the value of the comparison vibrator $d_{33}^{\left(2\right)}$ is given, $V_{1}$ and $V_{2}$ can be measured, and the value of the measured vibrator $d_{33}^{\left(1\right)}$ can be obtained. If $V_{1}$ and $V_{2}$ are processed by an electronic circuit, the quasi-static value of the longitudinal piezoelectric strain constant $d_{33}$ of the measured vibrator can be obtained directly. The piezoelectric coefficient measuring instrument of $d_{33}$ developed by (Foshan Zhuo Film Technology Co., Foshan, China) is shown in Figure 6b, and the measured value of the longitudinal piezoelectric coefficient is directly read out through special computer software.

The dynamic testing system is mainly composed of three parts, as shown in Figure 7. First, the sensor is loaded by the loading device probe, and then positive and negative electrodes are formed on the two sides of the piezoelectric layer, respectively. Second, the generated charge is amplified by a charge amplifier (SAPE01). Finally, an oscilloscope (UTD2102CEX) collects and displays the variation characteristics of its output voltage.

## 4. Results and Discussion

### 4.1. EDS and XRD Analysis

In order to study the composition and crystal orientation of each film, we deposited samples in the same furnace as the ZnO piezoelectric film sensor. We analyzed the composition of NiCr thin films, ZnO thin films, and SiO_2_ films by an EDS. Figure 8 shows the EDS spectrum of these films.

Figure 8a shows that the mass fraction ratio of Ni and Cr in the NiCr film is 53:47, which is close to the relative atomic mass ratio of NiCr of 59:52; the atomic composition ratio of Ni and Cr is 50:50, which is close to the atomic ratio of NiCr of 1:1. Figure 8b shows that the mass fraction ratio of Zn and O in the ZnO film is 80:20, which is close to the relative atomic mass ratio of ZnO of 65:16; the atomic composition ratio of Zn and O is 50:50, which is close to the atomic ratio of ZnO of 1:1. Figure 8c shows that the mass fraction ratio of Si and O in the SiO_2_ film is 66:34, which is close to the relative atomic mass ratio of SiO_2_ of 65:32; the atomic composition ratio of Si and O is 33:67, which is close to the atomic ratio of SiO_2_ of 1:2. For the convenience of calculation, the above data are rounded. The experimental analysis results show clearly that the composition ratios of NiCr, ZnO, and SiO_2_ films meet the requirements.

Figure 9 shows the XRD patterns of the ZnO films at optimized process parameters condition (Table 2). It can be seen that the preferred ZnO (002) textured films has a strong (002) diffraction peak near 2θ = 34.4°, which is close to the one reported 2θ = 34.42° [43], confirming the development of a polycrystalline hexagonal-like structure. The hexagonal wurtzite structure of ZnO films is most common in magnetron sputtering deposition because the growth along the c-axis direction is generally promoted [44]. Furthermore, because the wurtzite structure is the only non-centrosymmetric crystal phase of the ZnO material, the piezoelectric property of ZnO is closely related to this structure. High c-axis-oriented ZnO thin films are desirable to maximize the piezoelectric response of the material [43,45]. To further verify this result, we need to further analyze thin film morphology.

### 4.2. SEM and AFM Analysis

The surface micromorphology of thin-film samples was observed by SEM (JEM-2100F); Figure 10 shows the SEM images of NiCr, ZnO, and SiO_2_ thin films. It shows that the surfaces of NiCr, ZnO, and SiO_2_ films are flat, dense, evenly distributed and free of apparent defects, which meets the standard requirements of the experiment.

The quality of the piezoelectric film sensor is affected by the surface roughness and morphology of the deposited ZnO film. Therefore, the surface micromorphology of the ZnO film samples prepared in the same furnace was analyzed by using SEM and AFM. Figure 11a shows the SEM cross-sectional view of the ZnO film samples. It clearly shows the columnar growth of the ZnO thin film, perpendicular to the surface, which indicates that the preferential orientation of the samples is along the c-axis. This also confirms that the XRD results discussed above are correct. Figure 11b shows the AFM diagram of the ZnO film; the sample scanning area was 2 μm × 2 μm; the sample thickness was 0.98 μm; the highest protrusion on the surface was 49.3 nm; the average surface roughness was 2.91 nm; the root mean square of roughness was 3.475 nm. The ZnO thin films have relatively smooth and uniform surface morphology, and the ZnO grains are distributed in an egg shape and columnar growth.

### 4.3. Static Characteristics of the Piezoelectric Film Sensor

#### 4.3.1. Longitudinal Piezoelectric Coefficient $d_{33}$ Test

In order to ensure that the measurement accuracy was closer to the actual value, repeated measurements of multiple samples were carried out. Five samples were considered for each condition, the piezoelectric coefficient measuring instrument measured five different spots of each sample and the average was calculated. The test results (with two decimal places) are shown in Table 3; the maximum standard deviation of the five samples was ±0.03 pC/N.

The arithmetic mean of the longitudinal piezoelectric coefficients $\bar{d_{33}}$ of five samples can be obtained from Table 3.
$$\bar{d_{33}}=\frac{Y_{d33}^{1}{}_{}+Y_{d33}^{2}+Y_{d33}^{3}+Y_{d33}^{4}+Y_{d33}^{5}}{n}=\frac{1.36+1.32+1.35+1.39+1.38}{5}=1.36\mathrm{pC}/N$$
where $Y_{d33}^{1}$, $Y_{d33}^{2}$, $Y_{d33}^{3}$, $Y_{d33}^{4}$, $Y_{d33}^{5}$ are the arithmetic mean of the measurements of samples 1 to 5, respectively.

The ZnO piezoelectric film sensor was fabricated on a GH4169 superalloy steel substrate under existing experimental conditions. From the measurement results, the average value of the longitudinal piezoelectric coefficient was 1.36 Pc/N, lower than the previously reported results of 1.5 pC/N–5.7 pC/N [46]. There may be influencing factors, such as different substrates, different thickness of ZnO films, the effect of the electrode material and buffer layer, different sputtering equipment, effects of testing equipment and testing methods [47,48,49]. However, although the test results are slightly lower than the fabrication results on silicon wafer, it also proves that its application to the fabrication of smart bolts is feasible.

#### 4.3.2. Static Characteristics

To carry out experimental research on the thin-film pressure sensor, we put the five sensors deposited in the same furnace under the loading conditions of 50 g, 100 g, 150 g, 200 g, and 250 g, respectively, and converted to gravity (0.49 N, 0.98 N, 1.47 N, 1.96 N, 2.45 N). We measured the output voltage V and recorded the collected data, as shown in Table 4.

The experimental data in Table 4 were linearly fitted by Origin software and the fitting results are shown in Figure 12. The static calibration results of the five sensors are basically the same. The output voltage V is approximately linear with the force F in the variation range of 0–2.45 N. The fitting equation is:(9)$$\begin{array}{l} V_{1}=19.86F_{1}-3.35\\ V_{2}=19.23F_{2}-3.26\\ V_{3}=19.85F_{3}-3.58\\ V_{4}=19.93F_{4}-2.38\\ V_{5}=19.84F_{5}-2.59 \end{array}$$
where $V_{1}$, $V_{2}$, $V_{3}$, $V_{4}$, $V_{5}$ are the voltages of the piezoelectric film sensor samples 1 to 5 under different loads; $F_{1}$, $F_{2}$, $F_{3}$, $F_{4}$, $F_{5}$ are the corresponding loads to piezoelectric film sensor samples 1 to 5.

#### 4.3.3. Sensitivity Calculation

According to the definition of sensor sensitivity, the sensitivity of the ZnO piezoelectric film sensor can be expressed as:$$S=\frac{\Delta V}{\Delta F}$$
where ΔV is the output voltage variation and ΔF is the variation of force.

Since the sensitivity is the slope of the linear fitting line, according to Equation (9), the sensitivities of the five piezoelectric film sensors are:$$\begin{array}{l} S_{1}=19.86\;\mathrm{mV}/N\\ S_{2}=19.23\;\mathrm{mV}/N\\ S_{3}=19.85\;\mathrm{mV}/N\\ S_{4}=19.93\;\mathrm{mV}/N\\ S_{5}=19.84\;\mathrm{mV}/N \end{array}$$

The average value is $\bar{S}=\frac{S_{1}+S_{2}+S_{3}+S_{4}+S_{5}}{5}=19.77\;\mathrm{mV}/N$.

The results show that the ZnO piezoelectric film sensor can measure the force in the range of 0–2.45 N. The average value of the sensitivity is lower than that of a MEMS system previously reported [50]. Because it was the first time a ZnO piezoelectric film sensor was fabricated on a GH4169 superalloy steel substrate, no reference to a similar system sensitivity was found. According to the design experience, the sensitivity of the samples can meet the requirements for fabricating smart bolts.

### 4.4. Dynamic Characteristics of the Piezoelectric Film Sensor

Sample 5 was selected for the dynamic test experiment according to the test results of the piezoelectric coefficient $d_{33}$ and sensitivity. Figure 7 shows the test device. When the load was applied by the force loading device, a charge was generated on the surface of the ZnO piezoelectric layer. Then, the test results were displayed on an oscilloscope.

At room temperature, the force loading equipment loaded 0.49 N, 0.98 N, 1.47 N, 1.96 N, and 2.45 N, respectively. We defined the sum of the absolute values of the forward maximum and the reverse minimum as $V_{\max}$, and $V_{\max}$ increased from 6.81 mV to 45.85 mV. The output voltage response curve of the piezoelectric thin-film sensor as shown in Figure 13 demonstrates that the output voltage increases with the increase of F in the range of 0–2.45 N.

### 4.5. Stability Analysis of the Piezoelectric Film Sensor

To carry out the repetitive exploration experiment on the piezoelectric film sensor, it was placed at room temperature and loaded with the same load six times. Record the collected voltage value and perform linear fitting, Table 4 shows the measurement values and Figure 14 shows the fitting results.

Figure 14 shows that the measurement results of the piezoelectric film sensor tested six times are very close, and the standard deviation under different loads can be obtained according to Formula (10) and Table 5.
(10)$$\sigma =\sqrt{\frac{\sum_{i=1}^{n}{\left(x_{i}-x\right)}^{2}}{n-1}}$$
where n is the actual measurement time, $x_{i}$ is the result of each test, x is the average of multiple measurements and σ is the standard deviation.

The repeatability error is:(11)$$\delta =\frac{\sigma }{x}\times 100\%=4.03\%$$

The experimental results show that the maximum repeatability error of the same piezoelectric film sensor under the same load is 4.03%, lower than 8% for piezoelectric sensors previously studied [35]. Therefore, the piezoelectric film sensor has good repeatability, and it is a candidate for commercial application.

## 5. Conclusions

In conclusion, a ZnO piezoelectric film sensor was developed on a GH4169 superalloy steel substrate successfully. Via SME, EDS, and XRD, we analyzed the deposited quality of NiCr, ZnO, and SiO_2_ films by magnetron sputtering technology. We showed that the film surface had a relatively smooth and uniform microcosmic morphology. Five groups of samples were tested repeatedly by a $d_{33}$ piezoelectric coefficient instrument, and the average value of the $d_{33}$ coefficients was 1.36 pC/N, which can meet the application requirements of a ZnO piezoelectric film sensor. The sensitivity of the piezoelectric film sensor between 0 and 2.45 N was 19.77 mV/N, obtained by testing the dynamic and static characteristics of five groups of samples, and the output voltage increased with the increase of F. The repeatability test of the piezoelectric film sensor showed that the maximum repeatability error of the same sensor under the same load was 4.03%, which has good reusability. This paper only demonstrated that a piezoelectric film sensor can be fabricated on a GH4169 superalloy steel substrate and monitor the pressure change. However, there are still many issues, such as the rationality of film deposition process, the effects of the testing system accuracy, and whether there is an effect of the film quality when replacing the stainless-steel mask plate. Based on this method, we will fabricate smart bolts and improve their existing issues in order to ensure their better application in the field of structural health monitoring.

## Figures and Tables

**Figure 1 micromachines-13-00390-f001:**
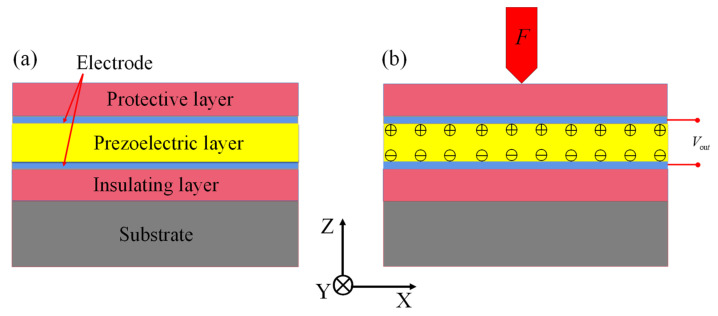
Working principle of piezoelectric film sensor: (**a**) normal state; (**b**) applied external force F.

**Figure 2 micromachines-13-00390-f002:**
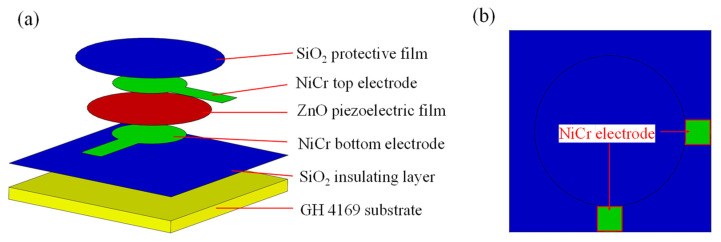
(**a**) Structural diagram of the piezoelectric film sensor; (**b**) structure of the leakage electrode.

**Figure 3 micromachines-13-00390-f003:**
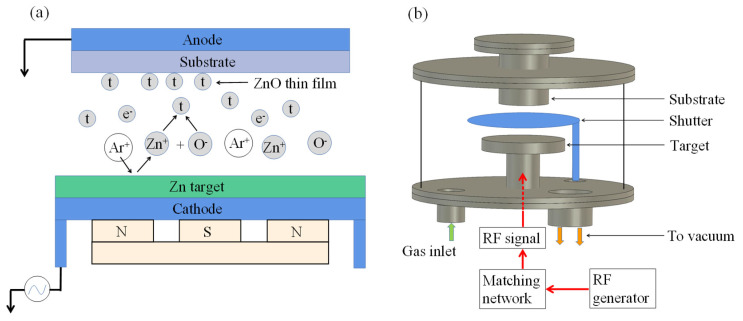
The principle (**a**) and equipment diagram (**b**) of RF magnetron sputtering.

**Figure 4 micromachines-13-00390-f004:**
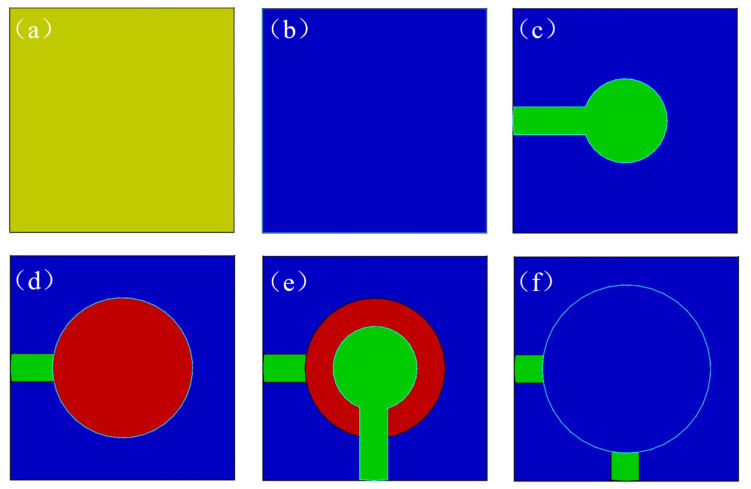
The primary fabrication process of the piezoelectric film sensor on GH4169 superalloy steel substrate: (**a**) polishing and cleaning of the substrate; (**b**) deposition of SiO_2_ insulating layer without a stainless-steel mask plate (SSMP); (**c**) preparation of NiCr bottom electrode film on SiO_2_ insulating layer; (**d**) the sputtering of the ZnO piezoelectric layer; (**e**) preparation of NiCr top electrode film on ZnO piezoelectric layer; (**f**) growing SiO_2_ protective layer film.

**Figure 5 micromachines-13-00390-f005:**
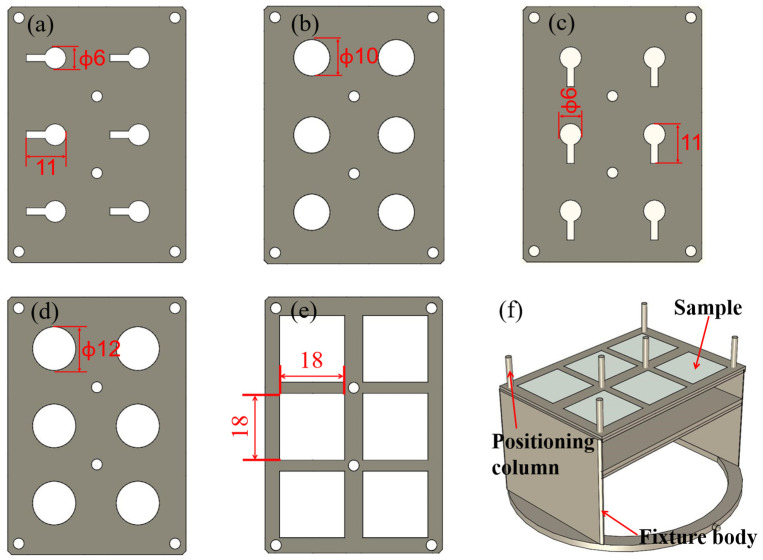
Stainless-steel mask plate of piezoelectric film sensor for fixture: (**a**) stainless-steel mask plate of NiCr bottom electrode; (**b**) stainless-steel mask plate of ZnO piezoelectric film; (**c**) stainless-steel mask plate of NiCr top electrode; (**d**) stainless-steel mask plate of SiO_2_ protective film; (**e**) sample positioning plates; (**f**) schematic diagram of fixture.

**Figure 6 micromachines-13-00390-f006:**
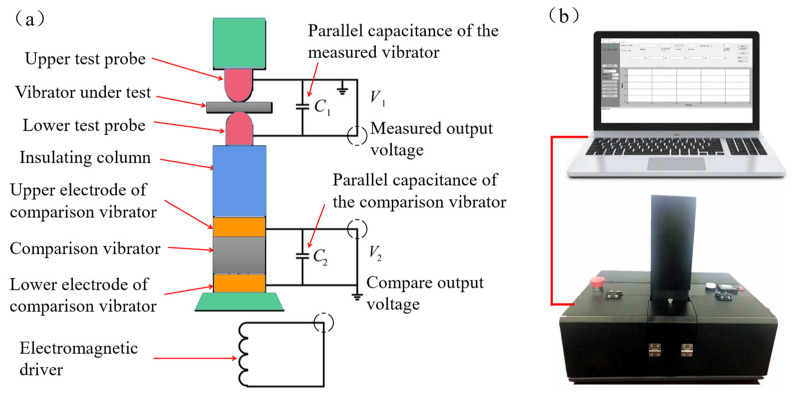
(**a**) Schematic diagram of quasi-static test method; (**b**) piezoelectric coefficient measuring instrument of $d_{33}$.

**Figure 7 micromachines-13-00390-f007:**
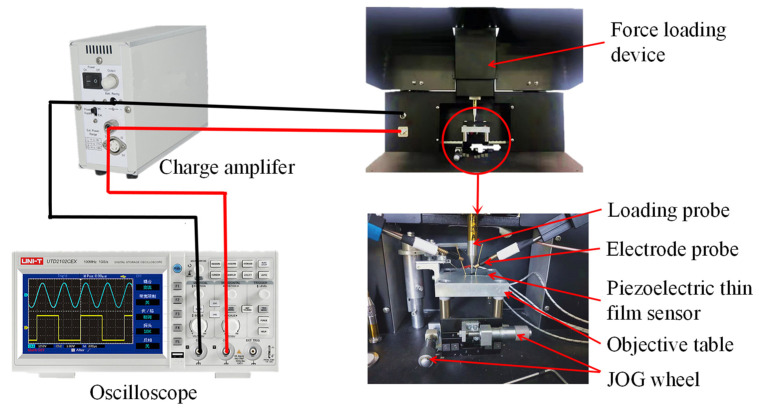
The dynamic testing system of the piezoelectric film sensor.

**Figure 8 micromachines-13-00390-f008:**
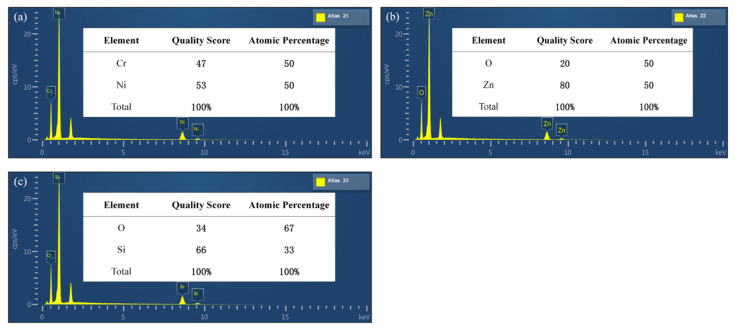
(**a**) Energy spectrum analysis of NiCr electrode film, (**b**) energy spectrum analysis of ZnO piezoelectric thin film, and (**c**) energy spectrum analysis of SiO_2_ insulating film.

**Figure 9 micromachines-13-00390-f009:**
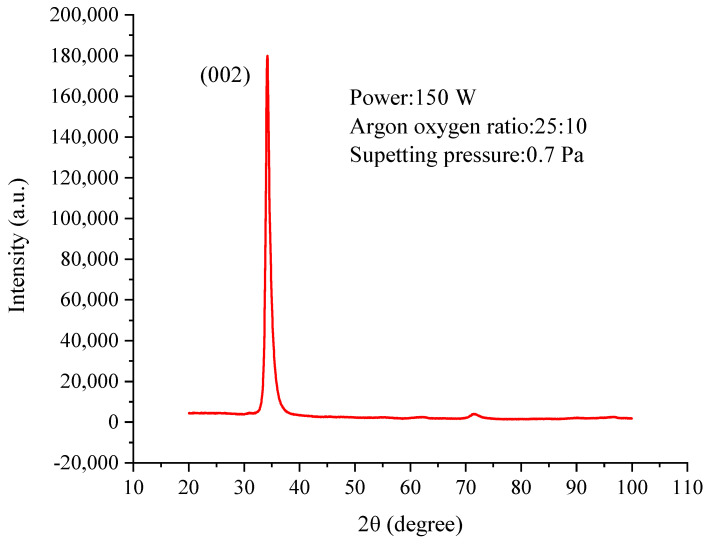
XRD spectrum of ZnO thin films.

**Figure 10 micromachines-13-00390-f010:**
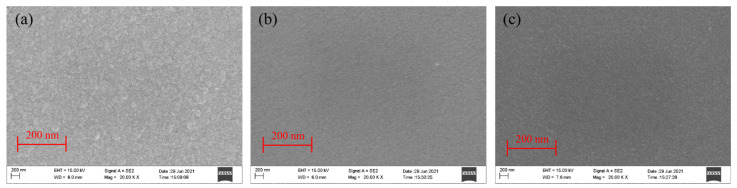
(**a**) SEM of NiCr electrode film surface, (**b**) SEM of ZnO piezoelectric film surface, (**c**) SEM of SiO_2_ insulating film surface.

**Figure 11 micromachines-13-00390-f011:**
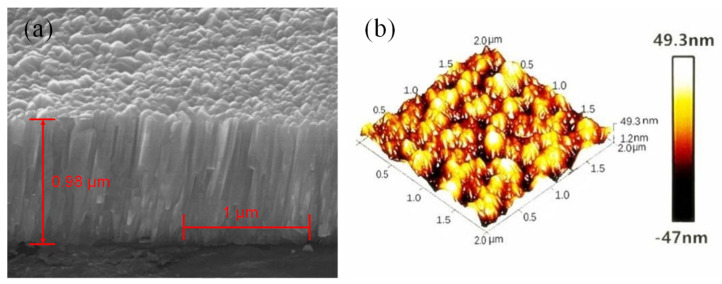
(**a**) SEM cross-sectional morphology and (**b**) AFM diagram of ZnO thin film.

**Figure 12 micromachines-13-00390-f012:**
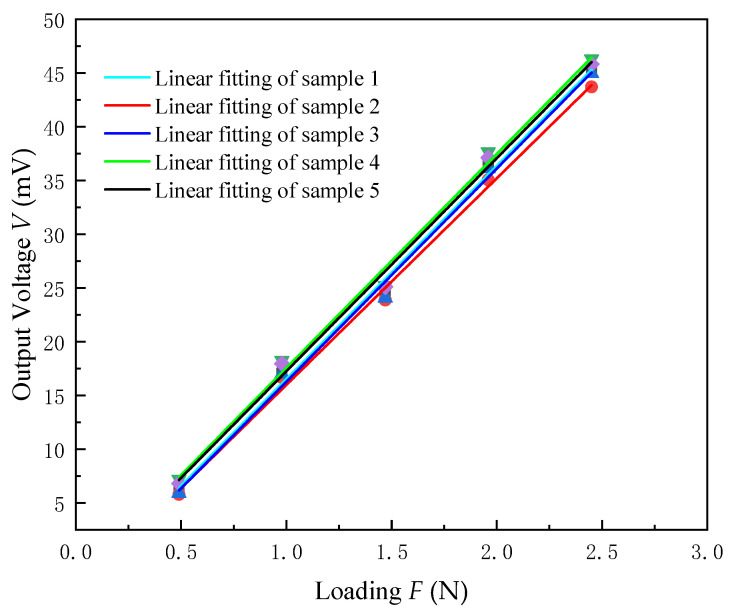
Voltage output characteristic curves of five piezoelectric film sensor samples under different pressures.

**Figure 13 micromachines-13-00390-f013:**
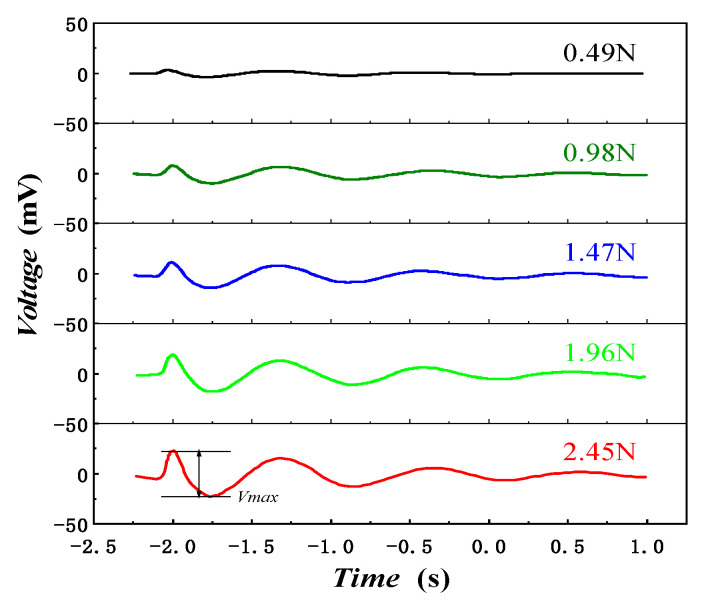
Output voltage response curve of piezoelectric film sensor.

**Figure 14 micromachines-13-00390-f014:**
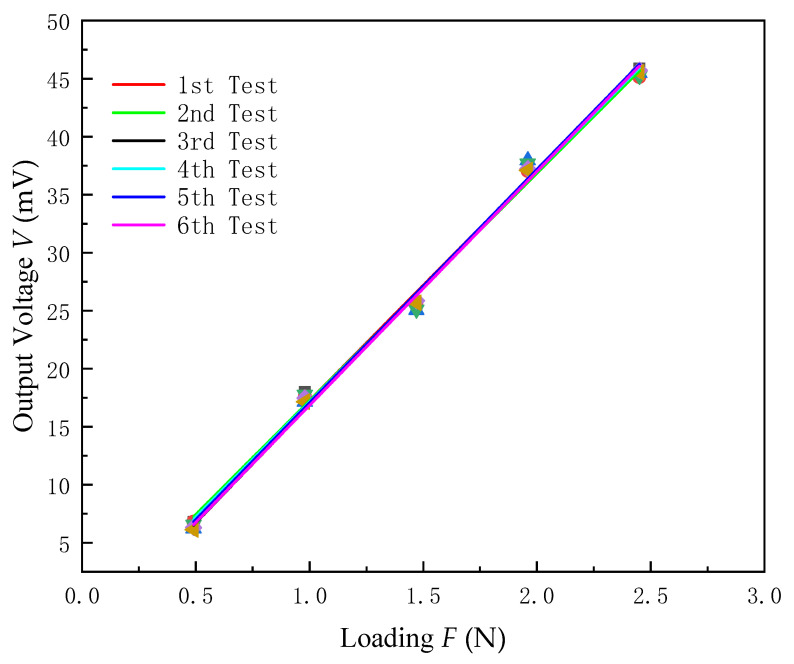
Output voltage of the same piezoelectric film sensor under identical conditions.

**Table 1 micromachines-13-00390-t001:** The properties of the GH4169 superalloy steel substrate.

Density (g/m^3^)	Young’s Modulus (GPa)	Thickness (mm)	Coefficient of Thermal Expansion (ppm/°C)	Melting Point (°C)
8.2	199.9	0.5	11.8 (20–100 °C)	1260–1340

**Table 2 micromachines-13-00390-t002:** Sputtering parameters of the thin films [41].

Film Layer	Background Vacuum (Pa)	Working Pressure (Pa)	Ar Flow (sccm)	O_2_ Flow (sccm)	Power (W)	Sputtering Time (min)
SiO_2_	1.0 × 10^−3^	0.7	20	10	250	60
NiCr	1.0 × 10^−4^	0.7	20	0	150	6
ZnO	1.0 × 10^−3^	0.7	25	10	150	80

**Table 3 micromachines-13-00390-t003:** Five piezoelectric film sensor samples were tested five times.

Sample	$1\mathrm{st}\;\mathrm{Test}\;\mathrm{Value}\;\mathrm{of}\;d_{33}\;\mathrm{pC}/N$	$2\mathrm{nd}\;\mathrm{Test}\;\mathrm{Value}\;\mathrm{of}\;d_{33}\;\mathrm{pC}/N$	$3\mathrm{rd}\;\mathrm{Test}\;\mathrm{Value}\;\mathrm{of}\;d_{33}\;\mathrm{pC}/N$	$4\mathrm{th}\;\mathrm{Test}\;\mathrm{Value}\;\mathrm{of}\;d_{33}\;\mathrm{pC}/N$	$5\mathrm{th}\;\mathrm{Test}\;\mathrm{Value}\;\mathrm{of}\;d_{33}\;\mathrm{pC}/N$	Average Value pC/N	STD
Sample 1	1.47	1.36	1.40	1.27	1.32	1.36	±0.03
Sample 2	1.21	1.37	1.28	1.35	1.40	1.32	±0.03
Sample 3	1.38	1.35	1.42	1.29	1.31	1.35	±0.02
Sample 4	1.39	1.43	1.37	1.35	1.42	1.39	±0.01
Sample 5	1.41	1.37	1.39	1.35	1.38	1.38	±0.01

**Table 4 micromachines-13-00390-t004:** Output voltage of five piezoelectric film sensors under five loads.

Imposed Loads (N)	Sample 1 Output Voltage (mV)	Sample 2 Output Voltage (mV)	Sample 3 Output Voltage (mV)	Sample 4 Output Voltage (mV)	Sample 5 Output Voltage (mV)	Average Value pC/N	STD
0.49	6.15	5.78	5.89	7.18	6.81	6.362	±0.27
0.98	17.07	16.67	16.98	18.25	17.97	17.388	±0.30
1.47	24.38	23.87	24.07	25.21	25.11	24.528	±0.27
1.96	36.56	35.06	36.10	37.65	37.13	36.500	±0.44
2.45	45.05	43.71	44.97	46.31	45.84	45.176	±0.44

**Table 5 micromachines-13-00390-t005:** Output voltage of the same sensor under identical conditions.

Loading F (N)	1st Test Output Voltage V (mV)	2nd Test Output Voltage V (mV)	3rd Test Output Voltage V (mV)	4th Test Output Voltage V (mV)	5th Test Output Voltage V (mV)	6th Test Output Voltage V (mV)	Average Value pC/N	STD
0.49	6.78	6.81	6.24	6.55	6.32	6.13	6.472	±0.13
0.98	17.98	17.60	17.15	17.78	17.46	17.14	17.518	±0.15
1.47	25.14	25.71	25.05	25.11	25.86	25.66	25.422	±0.16
1.96	37.23	37.03	37.97	37.68	37.24	37.11	37.377	±0.17
2.45	45.85	45.14	45.48	45.24	45.73	45.54	45.497	±0.12

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
