# Peer review of "Development and Characterization of ZnO Piezoelectric Thin Film Sensors on GH4169 Superalloy Steel Substrate by Magnetron Sputtering"

_micromachines, 2022, doi:10.3390/mi13030390_

Round 1

Reviewer 1 Report

Dear authors, I strongly appreciated the change you did on the introduction. Despite being a method-like paper rather than a scientific paper, I would accept the paper after some corrections/comments.

1. Line 12, a space is missing in between "isdeposited".
2. Line 22, replace mothed by method.
3. Is it important to mention positive for positive piezoelectric effect ? What would change if the response was negative ? Please, details.
4. The authors should describe more the figure legends, and I think especially to the figure 3.
5. Line 388, a "is" is missing.
6. The authors can defined what is "human factor" in line 350 ? This new paragraph needs references.
7. The paper contain too many figures and tables. I would suggest to switch some of them in a supplementary files, like for instance the table of deposition parameters.
8. Line 140, pi is of course a constant ...
9. Line 96 needs a reference.

Author Response

Dear Professor,

Thank you very much for your valuable comments on the paper.

Reviewer 2 Report

The authors have done sufficient improvement in the manuscript for passing publication criteria.

Author Response

Dear Professor,

Thank you very much for your valuable comments on the paper.

Yours sincerely

Guowei Mo

Reviewer 3 Report

I cannot decide whether it is suitable for publication

Author Response

Dear Professor,

Thank you very much for your valuable comments on the paper.

Yours sincerely

Guowei Mo

This manuscript is a resubmission of an earlier submission. The following is a list of the peer review reports and author responses from that submission.

Round 1

Reviewer 1 Report

The authors report on the growth and characterizations of a ZnO layer and devices. However, many similar systems are already existing and the message, the real input of this study is not clear. It looks like a collection of measurements and data without purpose. For this reason, I would not recommend it for publication in Micromachines.

1. Why do the authors used the word respectively in line 15 ?
2. The lines 17-18 should be rephrased.
3. Line 19, use “measurement error” instead of two repetition.
4. The technology of thin film is well advanced, as the authors suggest in the introduction. Saying that thin film instead of adhesive inclusion of the piezoelectric material will help is not what I called respond to a problematic... The abstract and the introduction should define the purpose of the study in a clear way.
5. Line 27, the authors should say more and not do “material dropping” like this.
6. Some references are needed in line 57-58.
7. There are too much back and forth in the introduction. Please make it shorter and more linear.
8. Line 142 means nothing. It's not due to the (5) equation that d is constant. Please, rephrase.
9. Line 152 needs reference.
10. Figure 5 and 6 are probably useless. And 6 is actually difficult to read. Please, place the lateral dimensions of your systems on the figure 4.
11. Experiment details of the characterizations should be placed in the correct paragraph.
12. A negative intensity on the XRD diffraction measurement means nothing.
13. Line 292, XRD absolutely do not say if piezoelectric properties are good or bad. Be careful.
14. Line 365, standard means room temperature?
15. Line 295, microcosmic means nothing.
16. Did the authors tried different growth parameters to increase the piezoelectric coefficient?
17. No comparison with state of the art have been done. Thw authors should compare with the existing piezoelectric material.
18. The authors said "for the first time", it could mislead the readers since piezoelectric ZnO have been deposited thousands of time. I agree that the substrate is new but what it brings to the story ? What are the advantages ?
19. The authors should say why the substrate is important. And I don't know if it can play a role since the buffer layers are huge.
20. The authors should compare the repeatability with other existing systems. What is the input of this study ?
21. Did the authors tried different device sizes ? What would it change ?
21. Did the authors tried different ZnO thicknesses ? What would it change ?

Reviewer 2 Report

In this manuscript, the authors have reported the development of a ZnO based piezoelectric based film sensors on alloy steel. After systhesis various characterization techniques have been used and the results have been reported. Although, many characterization techniqueshave been used and the results have been reported but there are some major lackings in the manuscript.

1: There are many mistakes of punctuation marks, spaces, commas, full stops etc. English review is thus needed

2: Some grammer corrections are required

3: Some sentences are very difficult to understand due to above mentioned reasons, as well as complex sentence structuring

4: It is highly recommended to use the full form of abbreviation in the main text...This should be done when the abrreviation is used for the very first time only.

5: The first part of the introduction needs a better writing and improvement in relation to the development of a story for the readers

6: The authors have discussed about the adhesion of the piezoelectric layers and have also claimed that this type of films have not been deposited before on the used alloy substrate. Did you measure the adhesion? I think it is important to support the novelity of the work with film-substrate adhesion measurements.

7: The only novelity is the one that has been reported in the line 98/99. I think this part of novelity should be more clearified

8: Throughout the article, many references are missing, especially for the equations and the experimental sections (e.g. line 150 to 153). Please recheck same for the results section

9: The authors discribe a single thing many times. Many sentences have been repeated multiple times. This should be corrected.

10: The floe of the gases should be mentioned in sccm. This is the more accurate representation.

11: It was difficult to understand the terms, "air pressure" and "background vacuum". Please recheck if this is the most common way. Does background means base pressure? or working pressure?

12: The deposition conditions should be explained in more details e.g. substrate bias and working pressure etc. Also, a schema of the deposition chamber, arrangement of the targets and substrates, decription of the purity of the substrates, would be a nice addition to the experimental section. 

13: Term sputtering time is used in the SiO2 table while, deposition time is used for NiCr. Both of these term mean different things. Sputtering doesnot always mean deposition.

14: Between the deposition of different layers, targets were replaced. This means that the samples were exposed to normal air. What do you think about theeffects of this exposure on quality of the developed coatings and also the interfacial strength betwwen the layers. A major drawback of this exposure can be the role of adsorbed oxygen on the surface of the electrodes and its interaction with piezoelectric material. This can in turn affect the voltage measurements. A better explanation of the effects/procedure to avoid such exposure is needed.

16: One approach to remove such exposure/oxygen exposure effects can be etching of already deposited layer, before next deposition (lets say etching of NiCr before ZnO deposition..and later ZnO etching before 2nd NiCr deposition). Was this carried out? if yes, it should be explained and the parameters are needed to be reported.

17: in section 3.1.3, the information on the thickness of the coatings is missing. I think it is importnant for the readers.

18: Al the tables in results section lack the standard deviation. Another issue is that the authors have reported 2 and in some cases 3 decimal places. I think the accuracy up to these decimal points is not possible. Especially for the EDS measurements. I suggest to reduce the decimal places.

19: symbol should be used for theta in XRD graph

20: Figure 11: The scale bar section is not clear. t is always nice to put the scales manually. can be easily done in power point. Also, the image quality is not so nice. If possible improve it.

21: the claim in line 306 about the planar growth based on cross.-section images is wrong. Also, in abstract, the word microstructure has been used. It should be structure. With XRD, microstructure cant be accessed.

22: I was looking to see the reasons in the results. One of the major issues which I felt while reading the manuscript was the lack of discussion. The section heading is results and discussion. This is more like a report where, the authors have just reported the results without any further explanation. I sugeest to provide some explanation fo the results for better understanding of the work. Also, a comparison with the literature results can be provided, where applicable.

4: 

Reviewer 3 Report

The submitted manuscript is well written. Due to my expertise, I am only able to assess the method of preparing piezoelectric sensors by magnetron sputtering. In this I have no reservations, only two minor points: in Table 1 the term "Air pressure" appears, which is an inappropriate term. I recommend using "working pressure", as in Tabs 2 and 3. Similarly, it is standard to use the units "sccm" when indicating the flow rate of individual gases instead of "ml/min" - Tables 1-3.
Thus, from my point of view, the article is suitable for publication almost in its present form.